# Development and Validation of the Jordanian Diabetic Health Literacy Questionnaire: Enhancing Diabetes Management in Arabic-Speaking Populations

**DOI:** 10.3390/healthcare12070801

**Published:** 2024-04-07

**Authors:** Walid Al-Qerem, Anan Jarab, Judith Eberhardt, Fawaz Alasmari, Safa M. Alkaee, Zein H. Alsabaa

**Affiliations:** 1Department of Pharmacy, Faculty of Pharmacy, Al-Zaytoonah University of Jordan, Amman 11733, Jordan; 201620088@std-zuj.edu.jo; 2College of Pharmacy, Al Ain University, Abu Dhabi 112612, United Arab Emirates; anan.jarab@aau.ac.ae; 3AAU Health and Biomedical Research Center, Al Ain University, Abu Dhabi 112612, United Arab Emirates; 4Department of Clinical Pharmacy, Faculty of Pharmacy, Jordan University of Science and Technology, Irbid 22110, Jordan; 5Department of Psychology, School of Social Sciences, Humanities and Law, Teesside University, Borough Road, Middlesbrough TS1 3BX, UK; j.eberhardt@tees.ac.uk; 6Department of Pharmacology and Toxicology, College of Pharmacy, King Saud University, Riyadh 12372, Saudi Arabia; ffalasmari@ksu.edu.sa; 7Department of Pharmacy, Faculty of Pharmacy, Petra University, Amman 11196, Jordan; 202020141@uopstd.edu.jo

**Keywords:** health literacy, Arabic, diabetes mellitus, validation

## Abstract

(1) Background: Amidst the global rise in type 2 diabetes mellitus (T2DM), effective management of the disease has become increasingly important. Health literacy, particularly in non-English speaking populations, plays a crucial role in this management. To address the lack of suitable tools for Arabic-speaking diabetic patients, this study developed and validated the Jordanian Diabetic Health Literacy Questionnaire (JDHLQ). (2) Methods: A sample of 400 diabetic patients from Jordan, with a balance in gender, age, and educational background, was recruited from an endocrinology outpatient clinic. The JDHLQ, consisting of informative and communicative sections, underwent rigorous validation. Utilizing principal component analysis and Rasch analysis, the JDHL’s reliability and validity were evaluated. (3) Results: The results showed moderate proficiency in understanding and communicating diabetes-related information and confirmed the reliability and validity of the JDHLQ. (4) Conclusions: These findings emphasize the importance of culturally appropriate health literacy tools in enhancing patient understanding, engagement, and overall management of T2DM in Arabic-speaking communities.

## 1. Introduction

The World Health Organization (WHO) describes diabetes mellitus as a chronic metabolic disorder characterized by elevated blood glucose levels. Deviations in insulin production, action, or both may be the cause of the disease [1].

The epidemiology of diabetes has changed significantly in the last three decades. In Jordan, specifically, the reported prevalence of type 2 diabetes mellitus (T2DM) in 2020 was 16%, and this is likely to reach around 21% in 2050 [2].

The challenges associated with T2DM and its potential for serious complications, such as blindness, kidney failure, heart attacks, strokes, and amputations of the lower extremities [3,4,5,6], highlight the vital role that health literacy plays in its management. Rigorous self-care practices are necessary for the effective management of T2DM. These include following recommended medication regimens, exercising, eating a healthy diet, and maintaining ideal blood glucose control [7,8]. In order to support patients in acquiring the necessary knowledge to manage their T2DM effectively, understanding the complexities of managing their condition, and enhancing their emotional health and self-efficacy, health literacy is essential. Better diabetes outcomes and an enhanced quality of life are achieved when patients with higher health literacy are better able to comprehend their condition, make educated decisions, communicate with healthcare providers, and follow their treatment plan [9,10].

Several tools have been used to evaluate health literacy in people with diabetes that encompass broader areas, such as critical and interactive health literacy abilities and functional features. When selecting a method for assessing diabetes health literacy, it is important to consider the instrument’s features, measurement scope, and suitability for the particular skills associated with managing diabetes in the Arabic context [11,12]. However, such tools are still lacking. Therefore, this study aimed to develop a tool in Arabic to measure diabetic patients’ health literacy. The application of this tool may optimize resource allocation, improve health outcomes, and enhance the quality of life for Arabic-speaking diabetic patients, providing benefits to both healthcare providers and patients.

## 2. Materials and Methods

The current study enrolled 400 diabetic patients attending the endocrinologist outpatient clinic at Albasheer Hospital between August and December 2023. The inclusion criteria were having a diagnosis with T2DM for at least 1 year, being 18 years old or above, and providing written informed consent to participate in the study. The files of diabetic patients who had a regular follow-up appointment the next day were reviewed and those who fulfilled the inclusion criteria were approached when they arrived at the clinic. Prior to enrollment in the study, participants were briefed on the study’s aims, assured that the information collected would be confidential, and informed that the patient could withdraw from the study at any point. In addition, the participants were informed that filling out the questionnaire would take about 10 min. Subsequently, all participants signed an informed consent form. This study followed the Declaration of Helsinki ethical guidelines. Ethical approval was secured from Al-Zaytoonah University of Jordan (Ref#18/09/2022–2023).

### 2.1. Data Collection and Study Instruments

The present study developed the “Jordanian Diabetic Health Literacy Questionnaire” (JDHLQ), whose design was based on a comprehensive literature review [13,14,15]. It was then subjected to a back-translation process into Arabic by two independent translators. The questionnaire included various sections and a data collection sheet designed to gather demographic information about the patients, with questions covering aspects such as gender, age, socioeconomic status, and education level. The first section of the questionnaire focused on the informative aspects of health literacy. It assessed patients’ abilities to evaluate, comprehend, and utilize information regarding T2DM. The second section, titled “communicative health literacy”, assessed the patient’s ability to effectively communicate about their diabetes. This included evaluating their capacity to explain the rationale behind a diabetic diet, articulate their own DM condition, and ask health professionals pertinent questions related to T2DM. The questionnaire included eight items on a 1–4 Likert scale, with five items in the informative section and three in the communicative section. A higher score on this scale indicated a greater ability, with the maximum achievable score being 32. Other information was retrieved from the patients’ files including HbA1c readings on the day of the visit and medication used.

### 2.2. Sample Size Calculation

The recommended approach for determining the required sample size for conducting a factor analysis in research is the participant-to-item ratio, with a suggested maximum ratio of 20:1 [16]. As the developed questionnaire included 8 items, the minimum required sample size was determined to be 160.

### 2.3. Tool Validation

The item development and content and face validity evaluation were performed by an expert panel consisting of two endocrinologists and a clinical pharmacist. A pilot study involving diabetic patients was conducted to confirm the questionnaire’s clarity and comprehensibility for Jordanian participants. The data from the pilot study were excluded from the statistical analysis. Furthermore, principal component analysis (PCA) was performed to evaluate the questionnaire’s construct validity, which was confirmed by performing confirmatory factor analysis (CFA). Cronbach’s alpha values were examined to ensure the internal consistency of each produced factor. Rasch analysis was performed to assess the ability of the tool to differentiate between patients’ abilities and assess the difficulty levels of the questionnaire items.

### 2.4. Statistical Analysis

The data analysis was conducted using the Statistical Package for the Social Sciences (SPSS) version 26 and R software version 4.3.3, specifically, the Test Analysis Modules (TAM) package version 4.1-4 and latent variable analysis (lavaan) version: 0.6-17. The suitability of the data for conducting PCA was assessed through Kaiser–Meyer–Olkin (KMO) analysis and Bartlett’s Test of Sphericity. The appropriate number of factors to be extracted was determined by examining the scree plot. A pattern matrix was generated using direct oblimin rotation. Communalities were evaluated, and any item with a communality below 0.3 was removed from the analysis. Additionally, factors loadings were evaluated and any item with loadings below 0.4 or with multiple loadings above 0.4 was dropped from the analysis. A multi-factorial Rasch analysis for polytomous responses was conducted. Person reliability and item separation reliability were computed to verify the suitability of the model. Additionally, infit/outfit statistics were produced. Infit and outfit mean square (MSQ) values ranging between 0.6 and 1.4 were considered acceptable [17]. Thresholds were computed to evaluate each item, and a Wright map was produced. Furthermore, confirmatory factor analysis (CFA) was applied to the suggested final model examined. To assess goodness of fit, χ^2^/df (minimum discrepancy), GFI (goodness-of-fit index), CFI (comparative fit index), standardized root mean squared residual (SRMR), (TLI) Tucker–Lewis index, and RMSEA (Root Mean Square Error of Approximation) were measured. Values ≤ 3 are acceptable for χ^2^/df [18,19]. RMSEA values ≤ 0.05 indicate a reasonable fit [20]. SRMR values ≤ 0.05 indicate an acceptable fit [21]. TLI values closer to 1 indicate a very good fit, while a value of 1 indicates a perfect fit [22]. GFI values equal to 1 indicate a perfect fit, values ≥ 0.95 indicate an excellent fit, and values ≥ 0.9 indicate an acceptable fit [18,23]. Similarly, CFI values of 1 indicate a perfect fit, values ≥ 0.95 indicate an excellent fit, and values ≥ 0.90 indicate an acceptable fit [24,25].

A bivariate analysis was conducted to compare the health literacy scores of different sample subgroups using the Mann–Whitney U test or the Kruskal–Wallis test. Significance was determined at *p* < 0.05.

## 3. Results

Table 1 displays the socio-demographic characteristics of a population of diabetic patients. The participant group displayed a median age of 58 years, with a majority being female, married, and with health insurance. The educational background varied among the participants: a significant portion had completed elementary school, followed by those who had completed high school, and a smaller fraction held college or university degrees. Most participants were married. In terms of monthly income, a substantial majority reported earning less than 500 Jordanian Dinars. The HbA1c median of the studied patients was above the normal level 8 (6.8–10). Only 6.5% of the studied sample were suffering from diabetic feet. The most used antidiabetic medication was metformin, followed by insulin.

On the JDHLQ, diabetic patients’ health literacy regarding diabetes-related knowledge was evaluated using a Likert scale ranging from one to four. Table 2 presents the frequency of responses to diabetes-related information and diabetes-related communication items. Participants rated their ability to understand various aspects of diabetes education. Remarkably, most participants displayed moderate proficiency, with most of the items having mean scores between 2 and 3. In particular, on the item “Understand the written information I receive from my healthcare provider”, 188 participants (47.0%) scored their ability as 3, while 74 (18.50%) rated themselves as 4, and the least was for the item “Evaluate the accuracy of diabetes-related information I obtain”, as only 13.5% gave themselves a rating of 4.

Examining the communicative domain, on the item “Explain why my diabetic diet is important”, only 49 (12.3%) gave themselves a rating of 4, while on the item “Ask health professionals a question”, a significant portion of 136 (34.3%) gave themselves a rating of 4.

KMO and Bartlett’s test of sphericity scree plots (Figure 1) identified two factors, informative and communicative health literacy. Table 3 presents scores, factor loadings, communalities, and Cronbach’s alpha for the JDHLQ items. For the informative aspect of health literacy, the analysis revealed strong internal consistency and reliability, as indicated by Cronbach’s alpha value. The mean score for this dimension suggested a moderate level of informative health literacy among participants. Similarly, the communicative health literacy dimension demonstrated a good level of internal consistency. The mean scores across this dimension pointed to a moderate ability among participants to communicate effectively about their diabetes.

The Rasch model indicated that the item separation and person reliabilities for the informative and communitive domains were (0.855, 0.804 and 0.798, 0.731 respectively). Infit and outfit MSQ values are presented in Table 4. The only item violating the acceptable range was “Evaluate the accuracy of diabetes-related information I obtain” (Outfit = 1.566 and Infit = 1.463). The thresholds displayed in Table 4 indicate that all the items had ordered response categories. The Wright map (Figure 2) confirms that the patients’ responses were distributed around all difficulty levels on both factors. The item thresholds were distributed among various difficulty levels, indicating different levels of challenge for the participants. The easiest items were the first threshold of items 7 and 8, while the most challenging item for participants to respond to was the last threshold of item 6.

CFA was conducted on the two-factor solution (8 items) to confirm model fit. The model yielded acceptable model fit indicators (χ^2^/df = 1.37, RMSEA = 0.03, SRMR = 0.02, GFI = 0.98, CFI = 0.99, and TLI = 0.993).

Significant differences were found in the health literacy scores between patients with different educational levels. Patients who had a college/university degree showed the highest diabetic health literacy scores, whereas patients who attended elementary school had the lowest score (medians = 24 vs. 20, respectively, *p* < 0.001). In addition, patients who earned more than 1000 JOD monthly had a significantly higher diabetic health literacy score compared with patients who earned less than 500 JOD monthly (medians = 27.5 vs. 21, respectively, *p* < 0.001). Furthermore, patients who had medical insurance showed lower diabetic health literacy scores compared with patients who did not (medians = 23 vs. 22, respectively, *p* = 0.012). Finally, patients who had diabetic feet had significantly lower diabetic health literacy scores (medians = 22 vs. 16, respectively, *p* < 0.001) (Table 5).

## 4. Discussion

The present study aimed to develop and validate a new tool, the Jordanian Diabetic Health Literacy Questionnaire (JDHLQ), a tool specifically designed to assess health literacy in Arabic-speaking diabetic patients. By focusing on this population, this study sought to bridge a critical gap in diabetes management and care. The JDHLQ was developed with the intention of providing healthcare professionals with a reliable and culturally sensitive instrument to better understand and address the health literacy needs of their patients, ultimately aiming to improve diabetes management outcomes in this demographic. The EFA suggested that a two-factor model was the best model for DHLQ, which was confirmed by the CFA.

The Rasch model validated the JDHLQ’s reliability, effectively measuring health literacy among Arabic-speaking diabetic patients. The exception, the item “Evaluate the accuracy of diabetes-related information I obtain”, which fell outside the ideal fit range, highlights the complexities in patients’ ability to assess information accuracy. This shows the need for enhanced education and clearer communication in diabetes management, especially in evaluating information accuracy. Research indicates that patients employ various methods to understand and manage their T2DM, including self-efficacy and medication adherence. However, gaps in T2DM education pose significant barriers to effective management [26]. This aligns with findings that diabetic patients have diverse information needs and often seek knowledge about treatment, disease progression, and self-management [27]. These findings emphasize the necessity for health literacy tools that support patients in accurately evaluating diabetes-related information, catering to their varied understandings and needs. Bridging the insights from the Rasch model with the observed moderate proficiency in health literacy among participants, it becomes evident that while the JDHLQ can measure literacy effectively, the actual understanding and communication capabilities of patients present a critical area for intervention.

The present study revealed that most participants demonstrated moderate proficiency in understanding and communicating diabetes-related information. The level of health literacy can significantly affect patients’ self-management of diabetes, including adherence to treatment and informed decision making. Comparing these results to existing research reveals gaps in patient knowledge and communication skills. Specifically, a systematic review in this area has highlighted gaps in patient and provider perspectives, including self-management dependent on patient knowledge, beliefs, attitude, and behavior, and poor interaction between patients and health providers, often due to language barriers and lack of communication skills [28]. Addressing these gaps through targeted educational interventions could lead to better health outcomes. This finding underscores the need for culturally tailored health literacy tools in diabetes management, as well as the need for patient education and support, especially in Arabic-speaking diabetic populations, to enhance overall T2DM care and management. Understanding the moderate proficiency in health literacy among participants sets the stage to explore further demographic factors influencing health literacy levels, revealing significant variations based on education, income, insurance status, and health complications.

The study findings indicated that DM patients with higher education levels and higher incomes have higher health literacy, which aligns with the results of the study conducted in Saudi Arabia [29]. Patients who did not have insurance were found to have higher health literacy compared to those who had health insurance, which could be explained by the fact that insured patients may be less motivated to know more about their medical condition since all their health-related expenses are covered. Moreover, diabetic patients who suffered from diabetic foot were found to have significantly lower health literacy scores, in line with findings from a study conducted in Pakistan [30], which suggested that poor health literacy is associated with macrovascular and microvascular complications. Recognizing these disparities in health literacy across different demographic groups underscores the critical need for innovative tools like the JDHLQ, tailored to meet the unique challenges faced by Arabic-speaking diabetic populations.

The JDHLQ stands as a vital advancement in health literacy assessment, particularly for Arabic-speaking diabetic populations. Its importance is accentuated when compared to other health literacy tools, which often lack cultural and linguistic tailoring. There is a need for health literacy tools that are sensitive to cultural and linguistic differences in diverse populations [31,32]. The JDHLQ, with its focus on Arabic-speaking individuals, addresses this gap by providing a tailored approach to assessing and enhancing health literacy specific to the cultural and linguistic nuances of this group. This makes it a more effective tool for identifying and addressing the unique health literacy needs of Arabic-speaking diabetic patients, thereby improving overall diabetes management in these communities.

### 4.1. Strengths and Limitations

The present study’s key strength lies in its innovative approach to addressing a significant gap in health literacy research, particularly for Arabic-speaking diabetic patients. By tailoring the JDHLQ specifically to this demographic, the research not only adds to the limited literature in this area but also provides a practical tool for healthcare practitioners in these communities. This specificity enhances the relevance and potential impact of the JDHLQ in improving diabetes management through better health literacy.

Furthermore, the comprehensive content and cultural sensitivity of the JDHLQ are significant strengths of this study. With the JDHLQ covering various health literacy aspects specifically tailored to diabetes management, its comprehensive approach ensures that the questionnaire addresses the multifaceted nature of diabetes care, from medication adherence to lifestyle changes. Additionally, the JDHLQ’s design, with a focus on cultural relevance for Arabic-speaking populations, has the potential to enhance its effectiveness. By considering cultural nuances and language specifics, the JDHLQ offers a valuable tool that is tailored to the needs and contexts of Arabic-speaking diabetic patients, making it a significant contribution to diabetes care and health literacy research.

However, this study is not without its limitations. Firstly, there is a potential lack of generalizability of the findings beyond the specific cultural and linguistic context of Jordan. This limitation points to a need for further research in diverse Arabic-speaking populations to validate the JDHLQ’s effectiveness more broadly. Additionally, potential limitations in sample size or diversity within the study population may impact the robustness of the findings.

There is the possibility of response bias in the current study, where participants may have offered socially desirable answers rather than true reflections of their health literacy. To counter this, future research could utilize anonymous surveys or indirect questioning techniques, thereby potentially enhancing the authenticity of the responses.

Considering further validation steps to improve the content validity of the JDHLQ tool, including measuring the correlation coefficient between the JDHLQ and a previously developed tool in Arabic, was not achievable due to the lack of previously published validated Arabic versions of the DHLQ.

In addition, the test–retest analysis measures the reliability and stability of the scores of a test obtained twice or more from the same individual [33]. Although this test could improve the reliability of the JDHLQ, it was not conducted due to the extensive length between each clinic visit by the same patient. Studies have shown that the most recommended time interval between tests in test–retest analysis is two weeks [34]. However, this cannot be implemented in the target population.

Finally, the variability in healthcare systems across different Arabic-speaking regions might affect the JDHLQ’s effectiveness. Comparative studies in these diverse healthcare settings could provide critical insights, enhancing the JDHLQ’s relevance and utility in varied contexts. Future research should also consider expanding the validation of the JDHLQ across different Arabic-speaking populations and settings. Such research could provide deeper insights into the nuances of health literacy across varied cultural contexts within the Arabic-speaking world.

### 4.2. Future Directions

Although beyond the scope of the present study, there is a clear need to explore the longitudinal impact of tailored health literacy interventions on diabetes management outcomes. This would promote an understanding of how improved health literacy affects not just immediate disease management but also long-term patient outcomes and healthcare costs. Furthermore, integrating and assessing the role of digital tools in health literacy strategies represents an exciting and relevant avenue for future research. This could include the development and evaluation of mobile health applications, online educational platforms, and other digital resources tailored to the specific needs of Arabic-speaking diabetic patients.

The future implications of health literacy research, particularly in the realm of chronic diseases like T2DM, are multifaceted. Emerging studies suggest a shift towards integrating digital technology, such as mobile health applications and online platforms, to enhance health literacy among diverse populations [35]. These technological advancements are expected to play a pivotal role in providing accessible, personalized, and interactive health education [36]. Furthermore, future research is likely to focus on the long-term outcomes of improved health literacy, examining its impact not only on disease management but also on overall healthcare costs and quality of life for patients [37,38]. This underscores the ongoing need for innovative approaches in health literacy research to adapt to changing healthcare landscapes and patient needs.

## 5. Conclusions

The present study developed and validated a new tool, the Jordanian Diabetic Health Literacy Questionnaire (JDHLQ), which marks a significant advancement toward understanding and enhancing health literacy among Arabic-speaking diabetic patients. The JDHLQ addresses a critical gap in diabetes management tools, particularly in non-English-speaking populations. The findings demonstrating moderate proficiency in diabetes-related knowledge and communication highlight the importance of tailored health literacy interventions. The JDHLQ’s potential to improve patient outcomes and quality of life emphasizes the need for ongoing research and adaptation of health literacy tools in diverse linguistic and cultural settings.

## Figures and Tables

**Figure 1 healthcare-12-00801-f001:**
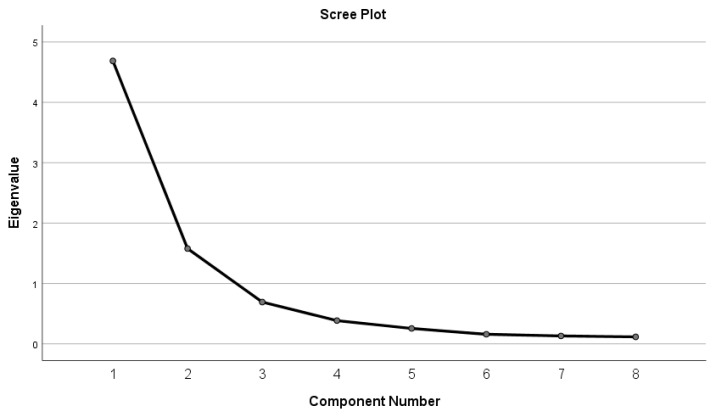
Scree plot of the factor analysis.

**Figure 2 healthcare-12-00801-f002:**
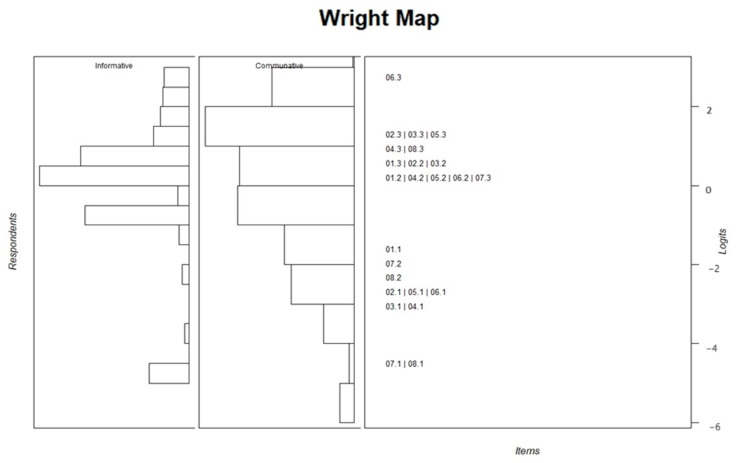
Wright map of the Rasch analysis.

**Table 1 healthcare-12-00801-t001:** Diabetic patients’ socio-demographic characteristics.

		Median (Percentile 25–75)	Count (%)
Age		58 (50–64)	
HbA1c		8.00 (6.80–10.00)	
Sex	Female		275 (68.8%)
	Male		125 (31.3%)
Education	Elementary		169 (42.5%)
	High school		142 (35.7%)
	College/university degree		87 (21.9%)
Marital status	Single		43 (10.8%)
	Married		355 (89.2%)
Monthly income	Less than JD 500		323 (81.2%)
	JD 500 or more		75 (18.8%)
Do you have health insurance?	No		84 (21.0%)
	Yes		316 (79.0%)
Medications	Insulin		150 (37.7%)
	Metformin		345 (86.7%)
	DPP-4 inhibitors		59 (14.8%)
	GLP-1-and dual GLP-1 GIP receptor agonists		15 (3.8%)
	SGLT2 inhibitors		12 (3%)
	Sulfonylureas		38 (9.5%)
	Thiazolidinediones (TZDs)		7 (1.8%)
Diabetic foot			26 (6.5%)

**Table 2 healthcare-12-00801-t002:** Frequency of responses to diabetes-related information and diabetes-related communication items.

Item	Frequency (%)			
	1	2	3	4
Informative domain				
Reading and understanding educational materials and booklets	44 (11.00%)	93 (23.30%)	198 (49.50%)	65 (16.30%)
Understand the written information I receive from my healthcare provider	41 (10.30%)	97 (24.30%)	188 (47.00%)	74 (18.50%)
Understand the information on diabetes management I obtain from the healthcare provider	42 (10.50%)	119 (29.80%)	174 (43.50%)	65 (16.30%)
Evaluate the accuracy of diabetes-related information I obtain	73 (18.30%)	151 (37.80%)	122 (30.50%)	54 (13.50%)
Understand the information I search for on diabetes	43 (10.80%)	108 (27.10%)	175 (43.80%)	74 (18.50%)
Communicative domain				
Explain why my diabetic diet is important	63 (15.80%)	155 (38.30%)	133 (33.30%)	49 (12.30%)
Explaining my diabetes condition to a healthcare provider	15 (3.80%)	80 (20.00%)	174(43.50%)	131 (32.80%)
Ask health professionals a question	15 (3.80%)	66 (16.30%)	183 (45.80%)	136 (34.30%)

**Table 3 healthcare-12-00801-t003:** Scores, factor loadings, communalities, and Cronbach’s alpha for JDHLQ items.

Question	Communality	Factor Loading	Mean (SD)	Cronbach’s Alpha	Total Mean (SD)
Informative health literacy					
Read and understand educational materials and booklets	0.608	0.747	2.71 (0.87)	0.831	13.19 (4.02)
Understand the written information I receive from my healthcare provider	0.753	0.844	2.74 (0.88)		
Understand the information on diabetes management I obtain from the healthcare provider	0.588	0.588	266 (0.87)		
Evaluate the accuracy of diabetes-related information I obtain	0.370	0.722	2.39 (0.94)		
Understand the information I search for on diabetes	0.759	0.859	2.70 (0.89)		
Communitive heath literacy					
Explain why my diabetic diet is important	0.549	0.926	2.42 (0.90)	0.811	8.57 (2.12)
Explain my diabetes condition to a healthcare provider	0.859	0.890	3.05 (0.82)		
Ask health professionals a question	0.798	0.741	3.10 (0.80)		

**Table 4 healthcare-12-00801-t004:** Outfits, infits, and thresholds of the JDHLQ items.

Item	Outfit MSQ	Infit MSQ	Thresholds		
			1	2	3
Informative domain					
Read and understand educational materials and booklets	1.1	1.113	−2.848	0.602	1.348
Understand the written information I receive from my healthcare provider	0.764	0.85	−2.952	0.454	1.301
Understand the information on diabetes management I obtain from the healthcare provider	0.973	1.034	−2.884	0.295	0.976
Evaluate the accuracy of diabetes-related information I obtain	1.566	1.463	−1.752	0.045	0.563
Understand the information I search for on diabetes	0.724	0.824	−2.842	0.322	1.137
Communicative domain					
Explain why my diabetic diet is important	1.238	1.214	−2.598	0.267	2.656
Explain my diabetes condition to a healthcare provider	0.707	0.784	−4.583	−1.839	0.132
Ask health professionals a question	0.866	0.879	−4.507	−2.195	1.052

**Table 5 healthcare-12-00801-t005:** Bivariate analysis of diabetic health literacy scores by different demographic characteristics.

		Median (Percentile 25–75)	*p*-Value
Sex	Female	22 (18–24)	0.068
	Male	23 (20–26)	
Education	Elementary school	20 (16–24)	<0.001
	High school	22 (19–24)	
	College/university degree	24 (22–30)	
Income	Less than JD 500	21 (18–24)	<0.001
	JD 500–1000	24 (21–28)	
	More than JD 1000	27.5 (23–32)	
Marital status	Single	23 (19–29)	0.175
	Married	22 (18–25)	
Insurance	No	23 (20–27)	0.012
	Yes	22 (18–24)	
Insulin	No	22 (18–25)	0.298
	Yes	22 (18–24)	
Metformin	No	23 (18–27)	0.182
	Yes	22 (18–25)	
DPP-4 inhibitors	No	22 (18–25)	0.185
	Yes	22 (18–27)	
GLP-1 and dual GLP-1 GIP receptor agonists	No	22 (18–25)	0.549
	Yes	21 (18–24)	
SGLT2 inhibitors	No	22 (18–25)	0.059
	Yes	26.5 (20–29)	
Sulfonylureas	No	22 (18–25)	0.305
	Yes	22.5 (19–27)	
Thiazolidinediones (TZDs)	No	22 (18–25)	0.003
	Yes	28 (26–30)	
Diabetic foot	No	22 (19–25)	<0.001
	Yes	16 (13–18)	

## Data Availability

The datasets analyzed for this study are available upon request.

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
