# Peer review of "Development and Validation of the Jordanian Diabetic Health Literacy Questionnaire: Enhancing Diabetes Management in Arabic-Speaking Populations"

_healthcare, 2024, doi:10.3390/healthcare12070801_

Round 1

Reviewer 1 Report

Comments and Suggestions for Authors

The authors of this manuscript developed and validated JDHLQ, as the culturally appropriate health literacy tool for Arabic-speaking diabetic patients, which is commendable.

However, certain parts of the manuscript need further clarification and/or correction.

In the Introduction section, given that the work does not refer to pediatric population, it is unnecessary to cite epidemiological data related to children (lines 40-42). In the sentence (lines 42-44), word "Jordan" should be mentioned only once, while the reference number 8 is not the best choice to support the statements mentioned in lines 48-50.

The Materials and Methods section is well written. It would be appreciated if the authors could provide information on how much time was allotted for filling out the questionnaire.

In the Results section, the data given in the text should be harmonized with those in the tables (table 2 - items and frequencies in line 147,148; lowest score in line 160; table 3 - factor loadings range in line 169; mean scores range in line 170). However, data in the text should be presented in a descriptive rather than in a numerical way (which is preferred in tables), to avoid the double listing / repetition of the same facts. Besides, regarding diabetes-related information, accuracy and reliability are not synonyms, and should not be used interchangeably. If possible, it would be very interesting to compare the JDHLQ scores of groups with different socio-demographic characteristics. Finally, it is recommended that the order of items should be the same in all tables. 

Regarding the Discussion section, the paragraph on the item "Evaluate the accuracy of diabetes-related information I obtain" should be more concise, and repetition should be avoided. Also, to enhance the readability and comprehension of the manuscript, it is suggested to refine the logical flow between and within paragraphs. On the other hand, strengths and limitations, as well as future directions, are well written.

Comments on the Quality of English Language

English language editing is suggested.

Author Response

-Thank you for comments and time that improved the quality of the manuscript

 the Introduction section:

  • given that the work does not refer to pediatric population, it is unnecessary to cite epidemiological data related to children (lines 40-42).

-This was corrected as suggested

  • In the sentence (lines 42-44), word "Jordan" should be mentioned only once.

-This was corrected as suggested

  • while the reference number 8 is not the best choice to support the statements mentioned in lines 48-50.

-The mentioned reference was deleted as suggested

The Materials and Methods section is well written. It would be appreciated if the authors could provide information on how much time was allotted for filling out the questionnaire.

-The approximate time to fill out the questionnaire was added to the Materials and Methods section.

In the Results section, the data given in the text should be harmonized with those in the tables (table 2 - items and frequencies in line 147,148; lowest score in line 160; table 3 - factor loadings range in line 169; mean scores range in line 170). However, data in the text should be presented in a descriptive rather than in a numerical way (which is preferred in tables), to avoid the double listing / repetition of the same facts. Besides, regarding diabetes-related information, accuracy and reliability are not synonyms, and should not be used interchangeably.

- the relevant parts of the Results section have been rewritten as follows:

“The participant group displayed a median age of 58 years, with a majority being female. The educational background varied among the participants: a significant portion had completed elementary school, followed by those who had completed high school, and a smaller fraction held college or university degrees. Most participants were married. In terms of monthly income, a substantial majority reported earning less than 500 Jordanian Dinars. Additionally, most participants were covered by health insurance.”

“For the informative aspect of health literacy, the analysis revealed a strong internal consistency and reliability, as indicated by the Cronbach’s alpha value. The mean score for this dimension suggested a moderate level of informative health literacy among participants. Similarly, the communicative health literacy dimension demonstrated a good level of internal consistency. The mean scores across this dimension pointed to a moderate ability among participants to communicate effectively about their diabetes.”

Furthermore, we have removed instances where ‘accuracy’ and ‘reliability’ were used interchangeably.

 If possible, it would be very interesting to compare the JDHLQ scores of groups with different socio-demographic characteristics.

-The results were added to the result section and further details about the impact of different sociodemographic variables were added as follows to the discussion:” The study findings indicated that DM patients with higher education levels and higher incomes have higher health literacy which aligns with the results of the study conducted in Saudi Arabia [31]. Patients who did not have insurance were found to have higher health literacy compared to those who had health insurance, which could be explained by the fact that insured patients may be less motivated to know more about their medical condition since all their health-related expenses are covered. Moreover, diabetic patients who suffered from diabetic foot were found to have significantly lower health literacy scores, in line with findings from a study conducted in Pakistan [32], which suggested that poor health literacy is associated with macrovascular and microvascular complications. Recognizing these disparities in health literacy across different demographic groups underscores the critical need for innovative tools like the JDHLQ, tailored to meet the unique challenges faced by Arabic-speaking diabetic populations.”

 Finally, it is recommended that the order of items should be the same in all tables. 

-This was corrected as suggested

Regarding the Discussion section, the paragraph on the item "Evaluate the accuracy of diabetes-related information I obtain" should be more concise, and repetition should be avoided. Also, to enhance the readability and comprehension of the manuscript, it is suggested to refine the logical flow between and within paragraphs. On the other hand, strengths and limitations, as well as future directions, are well written.

  • The paragraph has been rewritten as follows:

“The Rasch model validated the JDHLQ's reliability, effectively measuring health literacy among Arabic-speaking diabetic patients. The exception, the item "Evaluate the accuracy of diabetes-related information I obtain," which fell outside the ideal fit range, highlights the complexities in patients’ ability to assess information accuracy. This shows the need for enhanced education and clearer communication in diabetes management, especially in evaluating information accuracy. Research indicates that patients employ various methods to understand and manage their T2DM, including self-efficacy and medication adherence. However, gaps in T2DM education pose significant barriers to effective management [26]. This aligns with findings that diabetic patients have diverse information needs and often seek knowledge about treatment, disease progression, and self-management​​ [27]. These findings emphasize the necessity for health literacy tools that support patients in accurately evaluating diabetes-related information, catering to their varied understanding and needs.”

Additionally, the flow between paragraphs has been enhanced through the addition of the following connecting sentences:

“Bridging the insights from the Rasch model with the observed moderate proficiency in health literacy among participants, it becomes evident that while the JDHLQ can measure literacy effectively, the actual understanding and communication capabilities of patients present a critical area for intervention.”

“Understanding the moderate proficiency in health literacy among participants sets the stage to explore further demographic factors influencing health literacy levels, revealing significant variations based on education, income, insurance status, and health complications.”

“Recognizing these disparities in health literacy across different demographic groups underscores the critical need for innovative tools like the JDHLQ, tailored to meet the unique challenges faced by Arabic-speaking diabetic populations.”

Reviewer 2 Report

Comments and Suggestions for Authors

Sample selection in the research should be clarified. How were the 400 people included in the sample selected from the polyclinic? Only 5 questions were asked about the patients, and there is no information about diabetes management and adaptation to the disease.

How many people have insulin-dependent diabetes? Is there any patient diagnosed with diabetic foot? How were HbA1C values found? Do scale scores differ according to treatment methods? not questioned.

Some good and acceptable fit criteria should be given regarding the fit statistics used in structural equation modeling research.

1) χ2/df: Ratio of chi square and degrees of freedom;

2 RMSEA: Root Mean Square Error of Approximation;

3 SRMR: Standardized Root Mean Square Residual;

4 GFI: Goodness of fit index;

5 CFI: Comparative Fit Index

6 TL: Turker-Lewis Index

Exploratory factor analysis was performed, but Confirmatory Factor Analysis was not performed. The factors obtained in the Exploratory Factor Analysis analysis should be verified.

To investigate the content validity of the scale statistically, a standard scale, which has been previously developed in the field of interest and is accepted as a valid measure of the field of interest, and the newly developed scale must be applied to individuals at the same time, and the correlation coefficient must be calculated according to the scores individuals receive from both scales. Why was this method not preferred?

In the reliability section, only Cronbach's alpha coefficient was given and no test-retest analysis was performed.

Author Response

-Thank you for comments and time that improved the quality of the manuscript

Sample selection in the research should be clarified. How were the 400 people included in the sample selected from the polyclinic?

-The following was added to the manuscript: ”The inclusion criteria were having a diagnosis with T2DM for at least one year, being 18 years old or above, and providing written informed consent to participate in the study. The files of diabetic patients who had a regular follow-up appointment the next day were reviewed and those who fulfilled the inclusion criteria were approached when they arrived at the clinic. Prior to enrollment in the study, participants were briefed on the study aims, ensured that the information collected would be confidential, and that the patient could withdraw from the study at any point. In addition, the participants were informed that filling out the questionnaire would take about 10 minutes.”

Only 5 questions were asked about the patients, and there is no information about diabetes management and adaptation to the disease.

-further information about diabetes management was added to the results in addition to HbA1c scores and the presence of diabetic foot.

How many people have insulin-dependent diabetes?

-All patients enrolled in the present study had type 2 DM as it was an inclusion criterion. Information about patients who were using insulin were added to the manuscript.

 Is there any patient diagnosed with diabetic foot?

-This was added to the manuscript

How were HbA1C values found?

-It was retrieved from the patients’ files, this was added to the manuscript

 Do scale scores differ according to treatment methods? not questioned.

-The results were added to the result section and further details about the impact of different sociodemographic variables and treatment methods were added to the discussion: “The study findings indicated that DM patients with higher education levels and higher incomes have higher health literacy which aligns with the results of the study conducted in Saudi Arabia [31]. Patients who did not have insurance were found to have higher health literacy compared to those who had health insurance, which could be explained by the fact that insured patients may be less motivated to know more about their medical condition since all their health-related expenses are covered. Moreover, diabetic patients who suffered from diabetic foot were found to have significantly lower health literacy scores, in line with findings from a study conducted in Pakistan [32], which suggested that poor health literacy is associated with macrovascular and microvascular complications. Recognizing these disparities in health literacy across different demographic groups underscores the critical need for innovative tools like the JDHLQ, tailored to meet the unique challenges faced by Arabic-speaking diabetic populations.”

Some good and acceptable fit criteria should be given regarding the fit statistics used in structural equation modeling research.

  • χ2/df: Ratio of chi square and degrees of freedom;

     2 RMSEA: Root Mean Square Error of Approximation;

     3 SRMR: Standardized Root Mean Square Residual;

     4 GFI: Goodness of fit index;

    5 CFI: Comparative Fit Index

    6 TLI: Turker-Lewis Index

Exploratory factor analysis was performed, but Confirmatory Factor Analysis was not performed. The factors obtained in the Exploratory Factor Analysis analysis should be verified.

-The Confirmatory Factor Analysis was performed and model fit statistics were reported as suggested.

To investigate the content validity of the scale statistically, a standard scale, which has been previously developed in the field of interest and is accepted as a valid measure of the field of interest, and the newly developed scale must be applied to individuals at the same time, and the correlation coefficient must be calculated according to the scores individuals receive from both scales. Why was this method not preferred?

-The following was added to the limitation section: ”Considering further validation steps to improve the content validity of the JDHLQ tool, including measuring the correlation coefficient between the JDHLQ and a previously developed tool in Arabic was not achievable due to the lack of previously published validated Arabic versions of the DHLQ.”

In the reliability section, only Cronbach's alpha coefficient was given and no test-retest analysis was performed.

-The following was added to the limitation section “In addition, the test-retest analysis measures the reliability and stability of the scores of a test obtained twice or more from the same individual [33]. Although this test could improve the reliability of the JDHLQ, this was not conducted due to the extensive length between each clinic visit by the same patient. Studies have shown that the most recommended time interval between testing in test-retest analysis is two weeks [34]. However, this cannot be implemented in the target population.”

Round 2

Reviewer 2 Report

Comments and Suggestions for Authors

Corrections made by the author are sufficient.

In the table, p > 0.001 was written in education, income, and diabetic foot comparisons and in many places in the text. This article causes hesitation and confusion.

Should it be p<0.001?

Author Response

n the table, p > 0.001 was written in education, income, and diabetic foot comparisons and in many places in the text. This article causes hesitation and confusion.

Should it be p<0.001?

-Thank you for this. Sorry for this typo the signs were corrected.